# Photocatalytic Degradation of Chlorinated Hydrocarbons: The By-Product of the Petrochemical Industry Using Ag-Cu/Graphite Bimetallic Carbon Nitride

**Elsayed G. Blall** [1] , **Monica Toderas** [2,*] , **Abbas A. Ezzat** [1], **Hossam A. Abdou** [1], **Amira S. Mahmoud** [3] **and Fathy Shokry** [4]

1    Petrochemical Department, Faculty of Engineering, Pharos University, Alexandria 21526, Egypt; elsayed.gado.pt@pua.edu.eg or elsaiedblall@yahoo.com (E.G.B.); abbas.ezzat@pua.edu.eg (A.A.E.); hossam.anwer.pt@pua.edu.eg (H.A.A.)
2    Faculty of Sciences, University of Oradea, University St. No. 1, 410087 Oradea, Romania
3    Department of Environmental Studies, Institute of Graduate Studies and Research (IGSR), Alexandria University, Alexandria 21568, Egypt; amira.salah@alexu.edu.eg
4    Chemical Engineering Department, Faculty of Engineering, Portsaid University, Portsaid 42511, Egypt; fathi.shokry@eng.psu.edu.eg
*    Correspondence: atoderas@uoradea.ro

**Abstract:** In this study, the author improved and modified g-C$_3$N$_4$ by doping it with the metals Ag and Cu, which changed the photochemical properties of g-C$_3$N$_4$, narrowed the band gap, and improved the photocatalytic performance regarding quantum efficiency. Organic hydrocarbons such as 1,2-dichloroethane (DCE) are very stable prepared materials produced as intermediates to obtain polyvinyl chloride, and the prepared photo-catalyst is an innovative method for extreme decomposition of chlorinated hydrocarbons. However, some significant results were obtained using different analysis techniques. X-ray diffraction (XRD) and Fourier-transform infrared spectroscopy (FTIR) showed that the addition of Ag and Cu-NPS partially altered the structure of pure graphitic carbon nitride (g-C$_3$N$_4$-Pure). Scanning electron microscopy (TEM) analysis revealed that the morphological features of Ag-Cu/g-C$_3$N$_4$ contain quantum dots of Ag and Cu nanoparticles in addition to 2d-g-C$_3$N$_4$. The better separation of the photo-generated charge carriers is attributed to better photoactivity in the case of 0.3 g Ag-Cu/g-C$_3$N$_4$ with a reaction time of less than 30 min. Furthermore, the Ag-Cu/g-C$_3$N$_4$ recycling experiment showed that the catalyst remained stable after three stages of the pyrolysis experimental cycle. Another clear indicator of DCE degradation is the measurement using the titration of the Cl ions released by the decomposition.

**Keywords:** 1,2-dichloroethane (DCE); photocatalytic degradation: g-C$_3$N$_4$; Ag-Cu/g-C$_3$N$_4$; waste liquid; organic pollutants; photo-generated charge

## 1. Introduction

Photocatalytic strategies are complicated because many impartial influential parameters, collectively with initial pollutant concentration, catalyst loading, pH, response time, and dissolved oxygen, will impact the degradation of overall performance due to the tool's response [1]. DCE is one of the most critical chlorinated unstable natural compounds and is extensively utilized in industries as solvent, dry cleaner, degreaser, and chemical intermediate inside the manufacturing of artificial resins, plastics, and pharmaceuticals [2]. It is a synthetic chemical that not always located clearly inside the environment. It is a toxic, unstable, flammable, colorless liquid with a chloroform-like odor [3]. Furthermore, it was found to be a possible human carcinogen due to its conversion into chloro-cetaldehyde, which is considered to have mutagenic properties and has been indexed as a concern. A variety of methods, which include physical, chemical, and biological, have been studied for DCE degradation [4–8]. Due to the high chemical stability and low biodegradability of

DCE, advanced oxidation processes (AOPs) have recently received much attention [9,10]. To our knowledge, there are no reports of using Ag-Cu/g-$C_3N_4$ for the photocatalytic degradation of organic wastes such as methyl orange, methyl blue, or bisphenol A. But, Yongsheng Fu et al. used a Ag/g-$C_3N_4$ catalyst to degredate methyl orange (MO), methylene blue (MB), and neutral dark yellow GL (NDY-GL) under visible light irradiation. Photo-irradiation was carried out using a 500 W xenon lamp with UV cut-off filters (JB450) to remove radiation below 420 nm completely and ensure illumination by visible light only [11]. Among the various types of semiconductor photo-catalysts, $TiO_2$ is the most widely used in photocatalysis and has high photocatalytic efficiency for AgNP on the surface of $TiO_2$ P25. The obtained hybrid material shows a photocatalytic activity consisting of the degradation of 45% of the MO after 150 min [12]. The photocatalytic activity (PCA) of CuO NPs was evaluated via the photo-degradation of textile dye reactive red 120 (RR120) and methyl orange (MO) under sunlight irradiation. The result depicts around 90 % and 95% decolorization efficiency at 60 min, respectively [13]. However, $TiO_2$ has two main problems: a limited photoabsorption range and low quantum efficiency, which limit the practical applications of $TiO_2$ [12–14]. Therefore, it remains difficult to search for a more efficient photo-catalyst. Ag@g-$C_3N_4$ nanospheres degrade methyl blue in visible light with an efficiency of 85%, 1.8 times higher than pure g-$C_3N_4$ [15]. The chemical composition of g-$C_3N_4$ is generally nonstoichiometric and is often represented as $C_3N_4 + xH_y$ ($0 < x < 0.6$, $0 < y < 2$) [16]. For simplicity, the family of graphitic carbon nitride with a C/N ratio between 0.65 and 0.75 is called g-$C_3N_4$, although not precisely defined [17]. Owing to the chemical stability and unique electronic band structure, the polymeric semiconductor g-$C_3N_4$, as a low-cost, stable, and metal-free visible-light-active photo-catalyst, has been widely used in water splitting, organic photosynthesis, and pollutant degradation [18–20]. However, g-$C_3N_4$ photo-catalysts synthesized using traditional thermal polycondensation methods have small specific surface areas and wideband gaps; these defects severely suppress the photocatalytic activity [18,19]. g-$C_3N_4$ has a typical graphite layer structure, so the nanostructured design can enlarge the specific surface area of g-$C_3N_4$ and increase the number of active sites. Research has shown that theoretical g-$C_3N_4$ nanosheets exhibit higher specific surface area up to 2500 $m^2$ $g^{-1}$ [21]. The synthesis of g-$C_3N_4$ nanomaterials can effectively overcome the disadvantages of bulk g-$C_3N_4$ materials, such as a small specific surface area, low mass-transfer effects, and a high photo-generated carrier recombination rate, to improve the photocatalytic efficiency [22]. The semiconductor coupling of g-$C_3N_4$ expands the visible light response range because of different band gaps between the semiconductor and g-$C_3N_4$. The heterojunction structure and close contact interface improved the photocatalytic performance [23]. g-$C_3N_4$ has a band gap equivalent to 2.7–2.8 eV. This band gap provides a flexible channel to achieve control of the HOMO (highest-occupied molecular orbital) and the LUMO (lowest-unoccupied molecular orbital) [23]. g-$C_3N_4$ has three forms: the s-triazine-bases hexagonal structure, the tri-s-triazine (heptazine)-based structure, and the s-triazine-based orthorhombic structure [24]. Among the modification methods, metal loading is an important method to improve the performance of g-$C_3N_4$. In addition, the utility of metal/g-$C_3N_4$ composites is not restricted to photocatalysis; however, it can be carried out to natural systems, biosensors, fungicides, and so on. There have been detailed reports on the preparation and catalytic activity of g-$C_3N_4$, but few reports on metal/g-$C_3N_4$ composites in particular. Herein, we summarize recent progress in metal/g-$C_3N_4$ composites from synthetic methods to structures to applications [24,25]. Metals play an important role in enhancing the catalytic activity of g-$C_3N_4$. Its structure contains nitrogen triangles with six long-pair electrons; this unique structure is available for metal loading [26]. Therefore, metal loading is a proper method of designing new metal-semiconductor composites and has also been employed to modify the electronic structure of g-$C_3N_4$ [27,28]. The insertion of metal atoms effectively improves the photo-generated carrier mobility of g-$C_3N_4$, narrows the band gap, and further expands the visible-light response range [29]. Semiconductor photocatalysis was reported as a favorable strategy for degrading organic contaminants because of their superior electronic structure, low cost, and

excellent physicochemical properties [30,31]. The mechanism behind the degradation of organic pollutants is the production of strong oxidative holes and reducible electrons. These electrons and holes further produce hydroxyl ($OH^{\bullet-}$) and superoxide ($O_2^{\bullet-}$) radicals, respectively, which favor the effective degradation of organic pollutants [32]. In the literature, $g$-$C_3N_4$-based composites are suitable for the degradation of both the liquid and gas phases of organic pollutants [33]. Due to its photochemical stability, fascinating electronic band structures, and effective light harvesting accompanied by suitable bandgap energy of 2.7 eV, graphitic carbon nitride ($g$-$C_3N_4$) has been considered a promising metal-free photocatalyst for solving the energy crisis and environmental problems such as degradation of organic pollutants, and $CO_2$ photoreduction [34–36]. At the end of the introduction, we present Table 1, which documents all the research and photocatalysispreviously conducted on the cracking of 1,2 dichloromethane [4,7,11,12,37–39].

**Table 1.** Comparison between the prepared material and the prepared previously reported works.

| Catalyst | Performance | Reference |
|---|---|---|
| Reductive biodegradation of 1,2-dichloroethane via methanogenic granular sludge in lab-scale UASB reactors | 1,2-DCE was converted mainly to ethane (65–80%), and the hydraulic retention time varies between 10 and 20 h. | [4] |
| Degradation of 1,2-dichloroethane in V/TiO$_2$ | Complete photocatalytic degradation of 1,2-DCE was achieved after 120 min using UV radiation. | [7] |
| Degradation of 1,2-dichloroethane in Fe/TiO$_2$ | The photocatalytic performance is a function of retention time, and it would have competitive adsorption on the active site of TiO$_2$ between water vapor and 1,2-DCE. | [11] |
| Degradation of 1,2-dichloroethane in immobilized PAni-TiO$_2$ | The photocatalytic degradation of 1,2-DCE was about 60%, 90%, and 95% after 120 min, 240 min, and 300 min, respectively. | [12] |
| Degradation of 1,2-dichloroethane in a simulated wastewater solution: a comprehensive study by photocatalysis using TiO$_2$ nanoparticles and zinc oxide | It was found that with the UV method, just 55% of 1,2-DCA was removed after 6 h under 40 W UV radiation, but with the H$_2$O$_2$/UV method, the removal reached 88% for a similar length of time and radiation intensity. | [37] |
| Degradation of 1,2−dichloroethane in UV-M lamp. UV-N/S$_2$O$_4^{2-}$ | Complete degradation of 1,2-DCE after 300 min of irradiation time. | [38] |
| Degradation of gaseous 1,2-dichloroethane using a hybrid cuprous oxide catalyst | Degradation efficiencies of 83.8 and 82.2%. | [39] |
| Prepared material: photocatalytic degradation of 1,2 dichloroethane using Ag-Cu/graphite bimetallic carbon nitride radiated using UV | Degradation efficiencies of 0.3 g/100 mL Ag-Cu/g-C$_3$N$_4$ with a reaction time of less than 30 min of 100% with stable material and good reused several times. | - |

## 2. Materials and Methods

### 2.1. Materials

All chemical compounds used in this project were of analytical reagent grade. The 1,2-dichloroethan (DCE) ($C_2H_4Cl_2$, ACS reagent $\geq$ 99%, Merck, Rahway, NJ, USA) changed into used because of the goal pollutant. Urea (>97% Sigma–Aldrich, St. Louis, MO, USA) changed into used to synthesize the $g$-$C_3N_4$ photo-catalyst. $Cu(NO_3)_2 \cdot 3H_2O$ (2% mg in 20 mL water) and silver nanopowder and urea (AR, Sino Pharm Chemical Reagent Co., Ltd., Shanghai, China). Nitric acid was obtained from Sigma–Aldrich.

### 2.2. Laboratory Preparation of the Waste Liquid 1,2 Dichloroethane (DCE)

We prepared more concentrations of DCE 8400, 25,200, and 50,400 ppm as a scale of polluted materials with a (0.84) density of 1,2 dichloroethane.

### 2.3. Synthesis of the g-C₃N₄ Photo-Catalyst

Graphitic carbon nitride (g-C₃N₄) was synthesized through the thermal_solvothermal processing of urea (10 g) supersaturated liquid in a crucible with a lid under ambient pressure in the air [40,41]. After drying at 80 °C, the urea was installed in a muffle furnace (Isotemp Programmable Muffle Furnace Series, Fisher Scientific, Hampton, NH, USA) and heated to 550 °C for three hours to finish the reaction. The yellow-colored product obtained was washed with nitric acid (0.1 mol L$^{-1}$) and distilled water to eliminate any residual alkaline species (ammonia) adsorbed at the patterned floor, after which it dried at 80 °C for twenty-four hours. The weight of the yellow-colored powders was 0.4 g. The product was sonicated with ultra-pure water and nitric acid (1:1) for 2 h; the suspended product was centrifuged, washed several times with a mixture of ethanol and distilled water, and dried at 80 °C in a muffle and then a desiccator until it reached room temperature.

### 2.4. Preparation of Ag-Cu/g-C₃N₄ Catalysts [42]

The as-prepared graphic carbon nitride was dispersed in 200 mL of water and stirred well for 10 min. Then, we took 0.01 M of AgNO₃, 0.01 of Cu(NO₃)₂·3H₂O, and 1 g of PVP as a surface modification and placed them in the solution of graphitic carbon nitride. The solution was stirred well for 18 h under room temperature. Bulk bimetallic Ag-Cu is transformed into nanoparticles after 18 h of stirring with sodium borohydride and is distributed through g-C₃N₄ molecule. After filtration and washing with ethanol and water, that solution was sonicated with ethanol and distilled water. The samples were centrifuged at 3000 rpm, then collected and dried at 80 °C under a vacuum for 12 h. The bimetallic catalysts were denoted as Ag-Cu/C₃N₄ (1:1) with a yield of 0.8 gm, as shown in Figure 1.

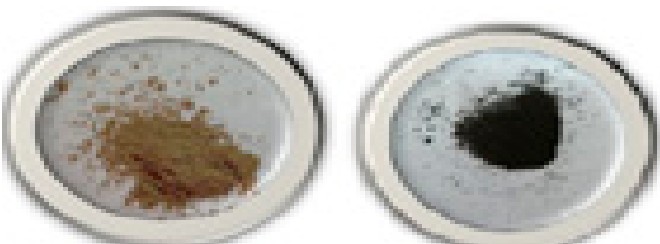

**Figure 1.** Picture of graphitic carbon nitride before and after doping with bimetallic silver and copper (Ag-Cu).

### 2.5. Characterization of g-C₃N₄ and Ag-Cu/g-C₃N₄ Catalysts

Fourier-transform-infrared (FT-IR) spectra were recorded on a Bruker VECTOR 22 spectrometer (Bruker, Mannheim, Germany) using the KBr pellet technique. The crystallinity and phase of Ag-Cu/g-C₃N₄ catalysts were characterized using X-ray diffraction (XRD) on a Bruker D8 Advanced diffractometer with Cu Kα radiation (λ = 1.5418 Å), and the scanning angle ranged from 10° to 100° of 2θ. The energy-dispersive X-ray Sem_SED_006, signal SED landing voltage of 20.0 kV WD 10.0 mm magnification ×450 and a vacuum-mode high-vacuum analysis was performed on both g-C₃N₄ and Ag-Cu/g-C₃N₄. High-resolution transmission electron microscopy (HR-TEM) micrographs were taken with a JEM-1400 Plus_2022 (Jeol Ltd., Tokyo, Japan).

### 2.6. The Photocatalytic Reactor Lab

Adsorption tests were conducted in a 500 mL PYREX® glass beaker (Glendale, AZ, USA) to assess the dynamic behavior for removing (DCE) under a (XE300-UV) Lamp Bulb, 300 W Xenon Short Arc, Fused Silica Glass (London, ON, Canada). To this end, the effects of dosage and initial concentration were pre-assessed on (DCE) removal. The adsorption kinetics of (DCE) were investigated at 25 °C using different weights of photocatalytic degradation (0.8, 0.5, 0.3, and 0.1 g) for Ag-Cu/g-C₃N4 at 1000 rpm stirring. The pH of the reaction solution was adjusted at 11 values by adding the NaOH solution to generate the OH$^{\cdot}$ radical, which helped to decompose the DCE [42].

The UV radiation tests were carried out on the radiation of (XE300-UV) Lamp Bulb, 300 W Xenon Short Arc, and used silica glass. Adsorption tests were completed in the 500 mL PYREX$^{®}$ glass beaker to evaluate the dynamic behavior for removing (DCE); see Figure 2. To this end, the effects of dosage and preliminary concentration were pre-assessed on (DCE) removal. The adsorption kinetics of (DCE) were investigated at 25 °C using the different weights of photocatalytic degradation (Ag-Cu/g-C$_3$N$_4$) (0.1, 0.3, 0.5, and 0.8 g/100 mL) at 1000 rpm of stirring. Three concentrations of the laboratory waste liquid (DCE) have been prepared (8400, 25,200, and 50,400 ppm). At first, the (DCE) solution was magnetically stirred without light irradiation for 30 min to obtain the adsorption_desorption equilibrium of (DCE) at the surface of the Ag-Cu/g-C$_3$N$_4$. After that, a simple (5 mL) measure was taken. We continued stirring at room temperature, then took a sample every 5 min and repeated this for 90 min. Firstly, the visible light must turn on, and the combined solution of the DCE and the Ag-Cu/g-C$_3$N$_4$ photo-catalyst becomes an exposed parallel under the visible light, followed by UV (300 W Xenon, UV lamp, $\lambda$ = 325 nm and the same steps are performed again. During the irradiation process, for every 5 min, about 5 mL aliquots of the samples were collected from the solution and then centrifuged at 3000 rpm to remove the contaminated catalyst. The change in the concentration of DCE molecules was measured from the absorbance of the degradable DCE samples using a Unicam-9423-UV-E spectrophotometer (Porto, Portugal) at its characteristic absorption peak wavelength of DCE at 550.6 nm. The photocatalytic degradation is illustrated in Figure 2.

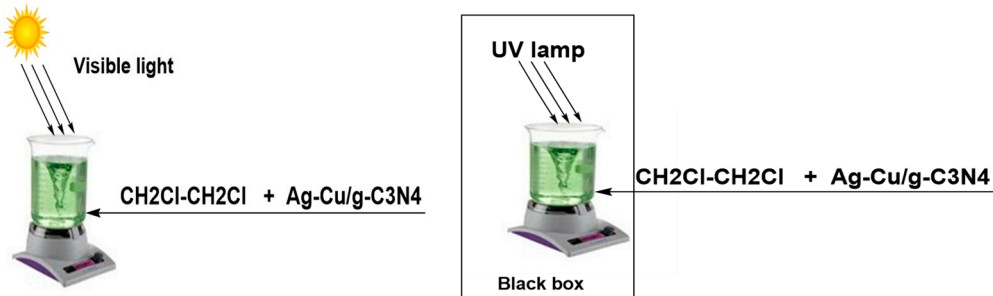

**Figure 2.** The photocatalytic reactor lab scale.

The Beer_Lambert law states that there can be a linear relationship between concentration and the absorbance of the solution, which allows the choice to be calculated through a manner of measuring its absorbance [43]. To display this linear dependence, four standard concentrations of DCE in water on an identical moderation were measured using the Unicam-9423-UV-E spectrophotometer at 556.6 nm. From the absorption spectra, a linear calibration curve of the absorbance, in place of attention, was created. This calibration curve can be used to determine the amount of unknown DCE solution by measuring its absorbance according to the Beer_Lambert law. The molar absorption coefficient ($\varepsilon M^{-1} cm^{-1}$) can be calculated. The photolysis test was conducted to determine the amount of DCE that degraded due to irradiation of 300 W Xenon, UV lamp, $\lambda$ = 325 nm for 60 min or more. For all tests, after 5 min, a sample was withdrawn to calculate the amount of DCE removal using the Beer_Lambert law, which is used in spectrometry and states that the absorbance of a species varies linearly with both the concentration of a solution C and the coefficient of extinction $\varepsilon$; when light passes through a distance, the path length is denoted by l, known as [43]:

$$As = \varepsilon \cdot Cl \tag{1}$$

The Langmuir–Hinshelwood (L–H) version is normally used to explain the kinetics of photo-degradation of natural pollution because the response usually happens among the adsorbed substrates at the catalyst surface and the photo-generated oxidants. The L–H kinetic Equation (3) can be expressed as follows [44]:

$$r = K_a \cdot K_r \cdot C/(1 + K_a C) \tag{2}$$

where r is the rate of photo-degradation, C is the DCE concentration at time t, kr is the charge constant, and $K_a$ is the adsorption equilibrium constant. The equation may be simplified to an obvious first-order Equation (4):

$$\ln (C/C_0) = K_a \cdot K_r \cdot t = K_0 \cdot t \tag{3}$$

When the completion of the reaction is confirmed by setting the absorption of DCE using the Unicam-9423-UV-E spectrophotometer at 556.6 nm and from the absorption spectra, the photo-catalyst is filtered, washed well with methanol, dried at 200 °C, and reused once more.

## 3. Results

### 3.1. FTIR Analysis

The FTIR spectra of the $g$-$C_3N_4$ and Ag-Cu/$g$-$C_3N_4$ catalysts with distinct Ag and Cu content materials are provided in Figure 3. All the absorption peaks of the Ag-Cu/$g$-$C_3N_4$ catalysts with different Ag and Cu content materials are almost the same as those of $g$-$C_3N_4$. The broad absorption peaks within 3215–3223 cm$^{-1}$ for $g$-$C_3N_4$ are attributed to the N-H stretching vibration and the O−H of the physically adsorbed water, respectively. The absorption peaks at approximately 1607, 1559, 1406, 1314, and 1235 cm$^{-1}$ can be ascribed to the typical stretching modes of CN heterocycles [45]. In addition, the sharp absorption peak at 808 cm$^{-1}$ is assigned to the characteristic breathing mode of tri-s-triazine units [45].

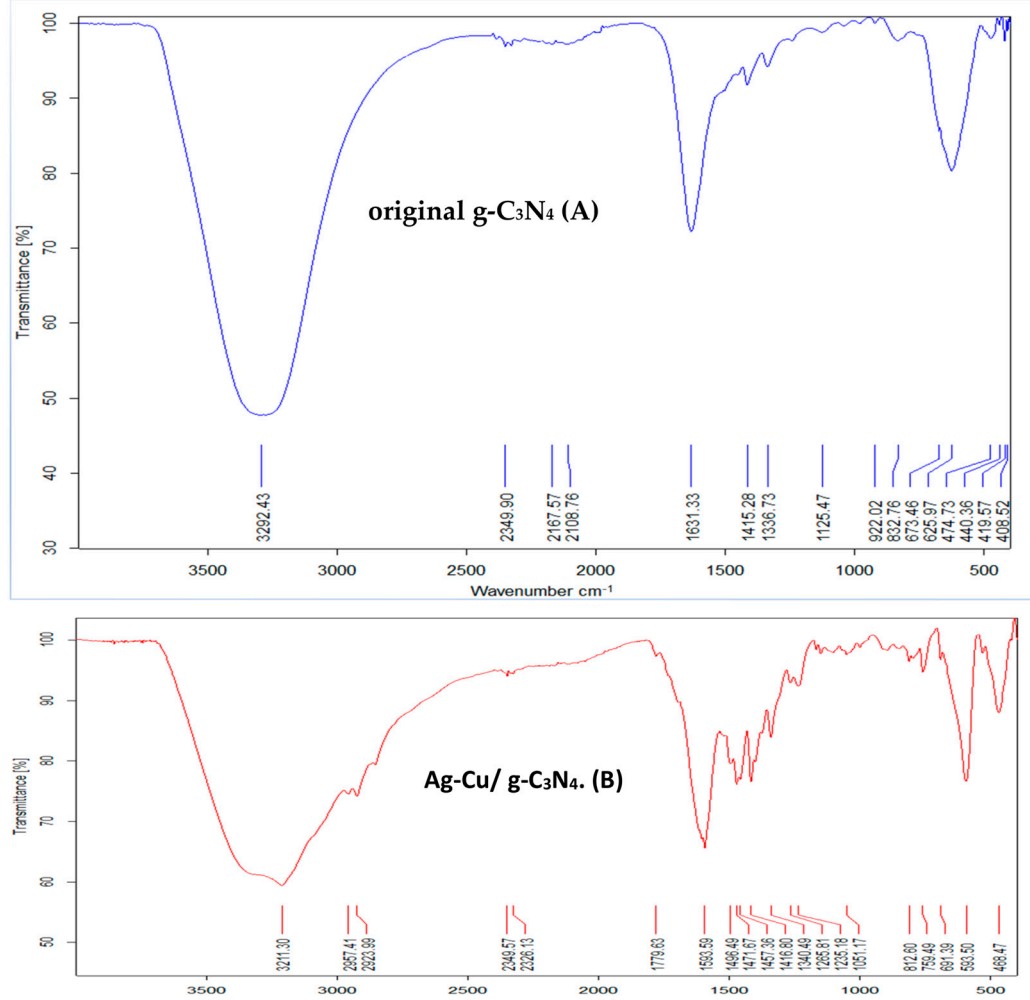

**Figure 3.** FTIR spectra of the original Ag-Cu/$g$-$C_3N_4$ and $g$-$C_3N_4$.

### 3.2. XRD Analysis

The crystallinity and degree of Ag-Cu/g-C$_3$N$_4$ catalysts were characterized via X-ray diffraction (XRD). For g-C$_3$N$_4$ and Ag-Cu/g-C$_3$N$_4$, from Figure 4a,b, there are diffraction peaks at 2θ of 13.0° and 27.5°, which may be listed to the (100) crystal plane bobbing up from the in-planar ordering of tris-triazine devices with a distance of 0.675 nm and a (002) crystal plane of the stacking of the conjugated fragrant device with an interplanar distance of 0.326 nm, respectively. For Ag-Cu/g-C$_3$N$_4$, there are observable peaks at 2θ of 38.1° and 44.3°, which may be assigned to the (111) and (200) crystal planes of the face-focused cubic shape of Ag, respectively; for Cu/g-C$_3$N$_4$, there are observable peaks at 2θ of 64.40 and 77°, which may be assigned to the crystal plane of the face-focused cubic shape of Cu, respectively. The intensity of the g-C$_3$N$_4$ diffraction peaks decreases with increasing Ag content. However, due to the relatively low Ag content, no clear Ag diffraction peak was observed for most samples of Ag and Cu [45].

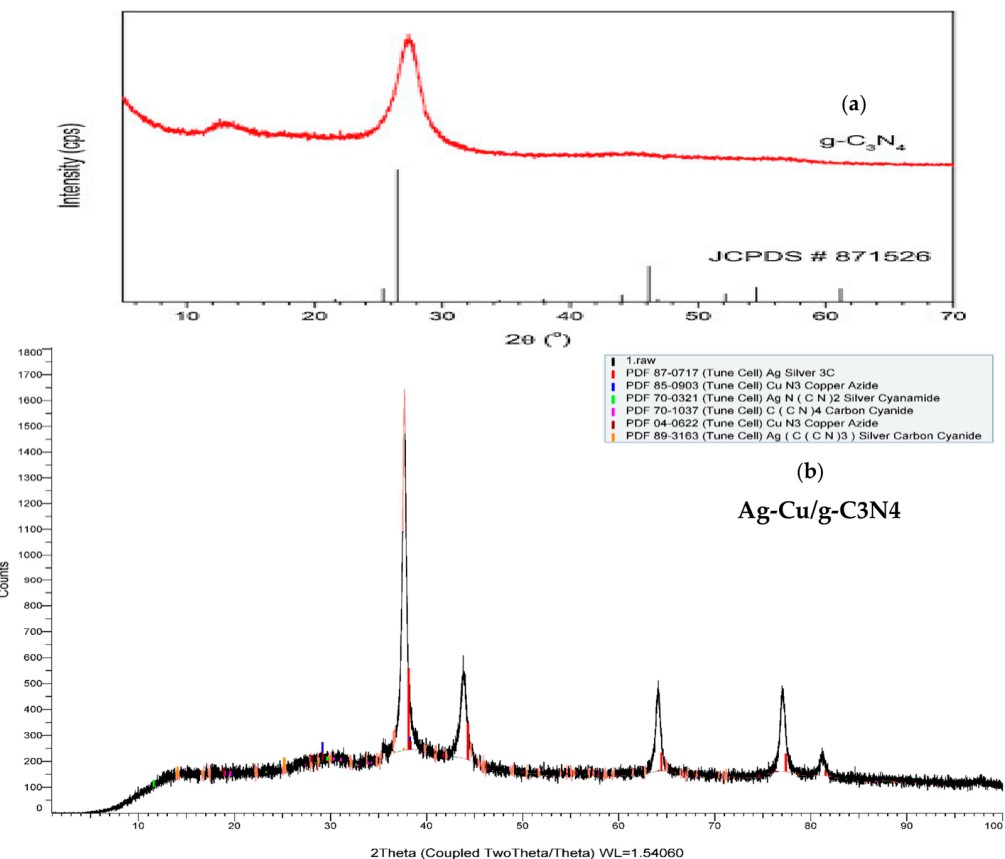

**Figure 4.** XRD diffraction patterns of (**a**) original C$_3$N$_4$ + x H$_y$ [40–42] and (**b**) Ag-Cu/C$_3$N$_4$ + x H$_y$ [45].

### 3.3. EDX and SEM Analysis

The energy-dispersive X-ray Sem_SED_006 in Figure 5a,b indicate the EDX evaluation of g-C$_3$N$_4$ with 29.69 ± 0.20 mass % for C and 70.31 ± 0.80 for N after adding two molecules of ammonia to the resin, which was confirmed due to the high alkalinity of resin, which appears at 3215–3223 cm$^{-1}$ at FTIR, while Ag-Cu/g-C$_3$N$_4$ illustrates the life of Ag, Cu, C, N, and O (33.45 ± 0.45, 20.26 ± 0.50, 11.02 ± 0.16, 19.46 ± 0.43, and 15.81 ± 0.47), respectively, as essential factors. The levels of the peaks are associated with the attention of every element. Also, the dispersion of those species in stated samples may be decided in dot mapping. The presence of oxygen with a mass of 15.81 ± 0.47 may be due to the presence of 3H$_2$O molecules resulting from the hydrated form or adsorption of oxygen.

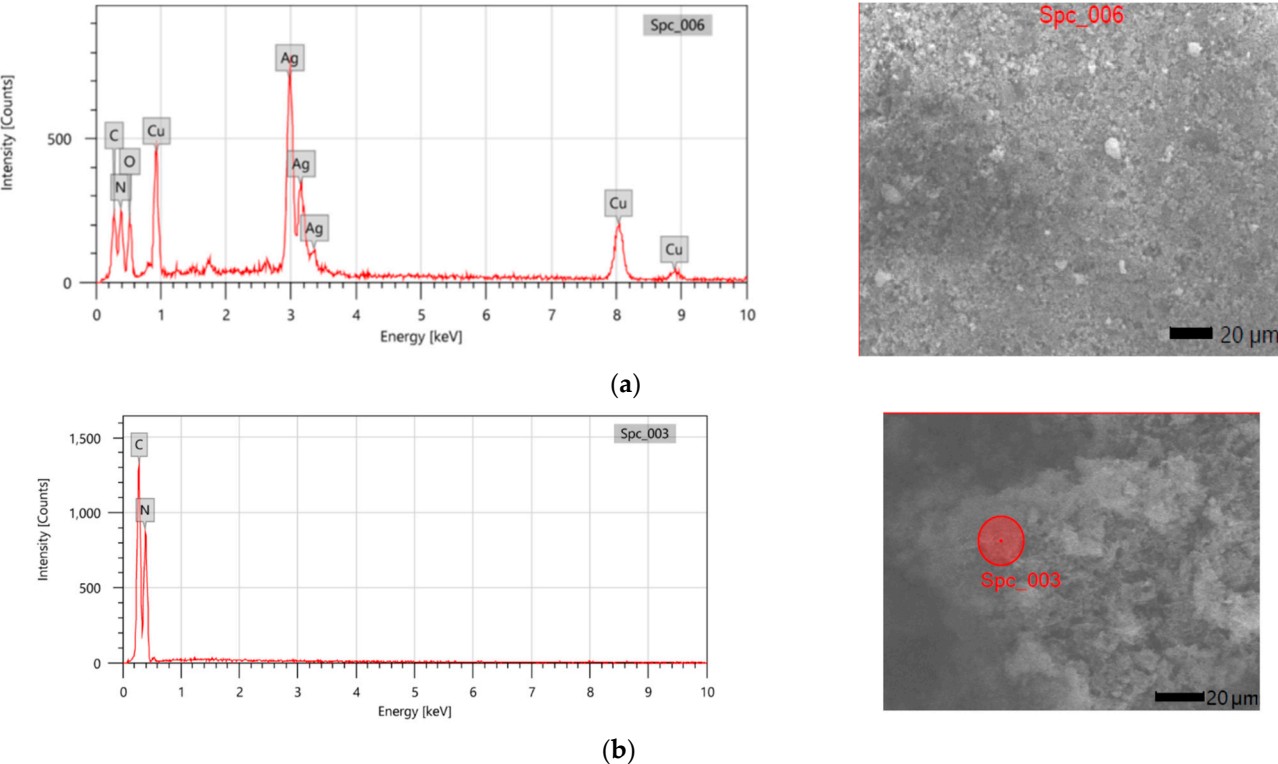

**Figure 5.** (**a**) Energy-dispersive X-ray (Sem-EDX) analysis of Ag-Cu/C$_3$N$_4$ and its elemental area mapping showing the distribution of elements. (**b**) Energy-dispersive X-ray (Sem-EDX) analysis of g-C$_3$N$_4$ and its elemental area mapping showing the distribution of elements.

The SEM analysis of both g-C$_3$N$_4$ and Ag-Cu/g-C$_3$N$_4$ confirms the presence of both molecules in the nano state, Figure 5 [46].

### 3.4. TEM Analysis

High-resolution transmission electron microscopy (HR-TEM) micrographs, Figure 6, show the two images (a and b) of the TEM of Ag-Cu/g-C$_3$N$_4$, and we notice that many dots have different nano sizes such as 7.87, 8.5, 10.64, 10.48, and 3.81, which may be attributed to quantum dots and 2D nanosheets, which confirmed the nanostructure of Ag-Cu/g-C$_3$N$_4$ [45]. This is attributed to the highly photocatalytic ability of Ag, Cu, and g-C$_3$N$_4$ due to the quantum dot particles and narrowband gap and to the high surface area of the pours catalyst due to the exit of many pours in the surface structure as seen in the (HR-TEM) image [47].

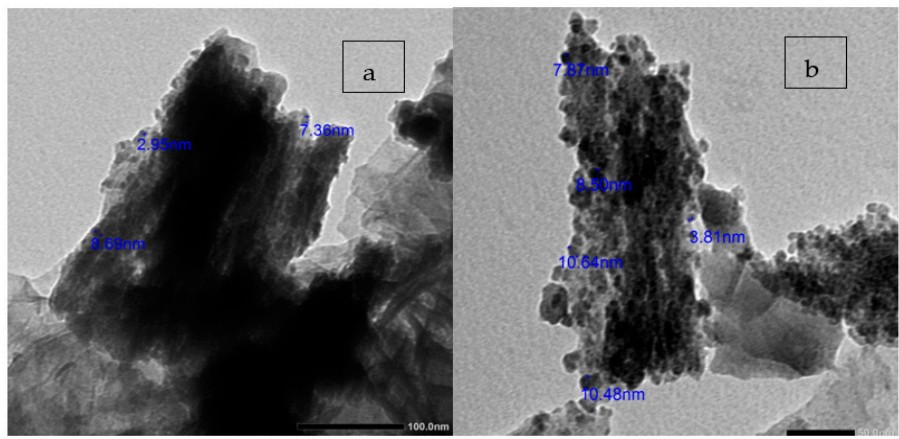

**Figure 6.** (**a**) and (**b**)TEM photos of Ag-Cu/C$_3$N$_4$.

### 3.5. Photocatalytic Degradation Activity of Ag-Cu/g-C$_3$N$_4$

The photocatalytic degradation activities of Ag-Cu/g-C$_3$N$_4$ at various gm/100 mL concentrations (0.1, 0.3, 0.5, and 0.8 g) were examined towards the photocatalytic degradation of DCE (8400, 25,200, and 50,400) ppm and also under similar experimental conditions for making a comparison.

### 3.5.1. Decomposing DCE under the Visible-Light Region

We used a 300 W Xe lamp (PLSSXE300C) to simulate solar light, which operates in continuous mode. First, the visible absorption spectra of the DCE (50,400 ppm) solution in the presence of different weights (0.1,0.3,0.5, and 0.8 g) of Ag-Cu/g-C$_3$N$_4$, Figure 7a,b, show no degradation with visible light irradiation but slight absorption and no liberated Cl ion detected via titration with AgNO$_3$, which is indicated only by the absorption of DCE on the service of a catalyst without degradation. This is clearly shown by the slow concentration of the resulting DCE shown in Figures 7 and 8 and Table 2 The result also shows that the wavelength of the light incident on the catalyst is smaller than the band gap of Ag, Cu nanoparticles (2.51 eV), (1.98–2.02 eV), and the band gap of g-C$_3$N$_4$ (2.7–2.8 eV), which slightly accelerates the decay of the narrowband gap in visible light in addition to silver and copper. Another element had to be introduced soon into the g-C$_3$N$_4$ content to degrade the DCE components in the visible spectrum [48,49].

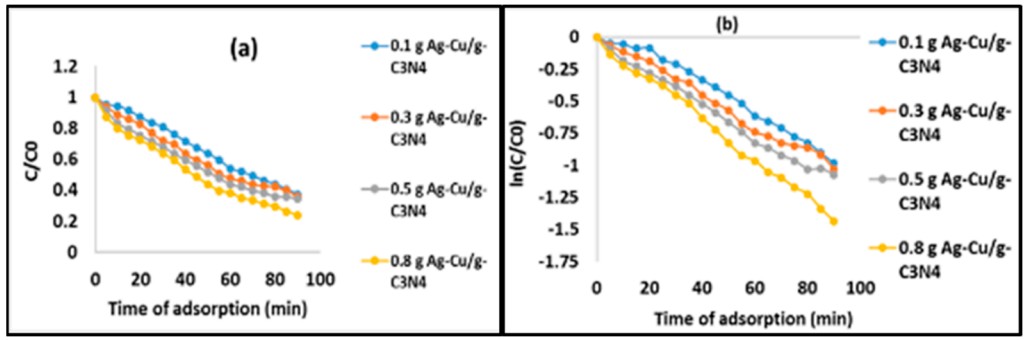

**Figure 7.** (**a**) Plot of (C/C$_0$) and (**b**) ln(C/C$_0$) against reaction time for photocatalytic adsorption of DCE with a different catalysts dose.

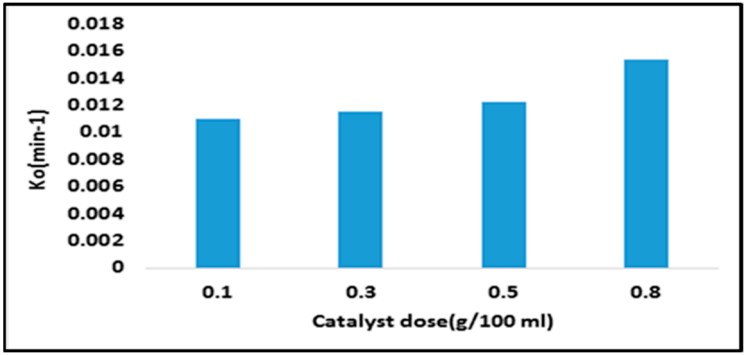

**Figure 8.** Rate of adsorption DCE on the surface of the catalyst using 60% DCE (504 g/100 mL) and 0.8 g Ag-Cu/g-C$_3$N$_4$.

**Table 2.** Pseudo-first-order apparent constant values for the 50,400 ppm DCE adsorption using different doses of Ag-Cu/g-C$_3$N$_4$.

| DCE (ppm) | $K_o$ (min$^{-1}$) | | | | $R^2$ | | | |
| --- | --- | --- | --- | --- | --- | --- | --- | --- |
| | Weight of Ag-Cu/g-C$_3$N$_4$/100 mL | | | | | | | |
| | 0.1 | 0.3 | 0.5 | 0.8 | 0.1 | 0.3 | 0.5 | 0.8 |
| 50,400 | 0.011 | 0.0115 | 0.0122 | 0.0154 | 0.9849 | 0.9914 | 0.9925 | 0.9957 |

### 3.5.2. Decomposing DCE under the UV-Light Region

It was found that Ag-Cu/g-C$_3$N$_4$ showed better degradation efficiency in the case of a low concentration of DCE (8400 ppm), taking a shorter time, as shown in Figure 9a–c. The blank experiment result indicates that the degradation of DCE can be neglected in the absence of catalysts and in a dark system. The DCE photocatalytic degradation reaction processes over different concentrations of Ag-Cu/g-C$_3$N$_4$ catalysts were successfully fitted with the pseudo-first-order kinetic model.

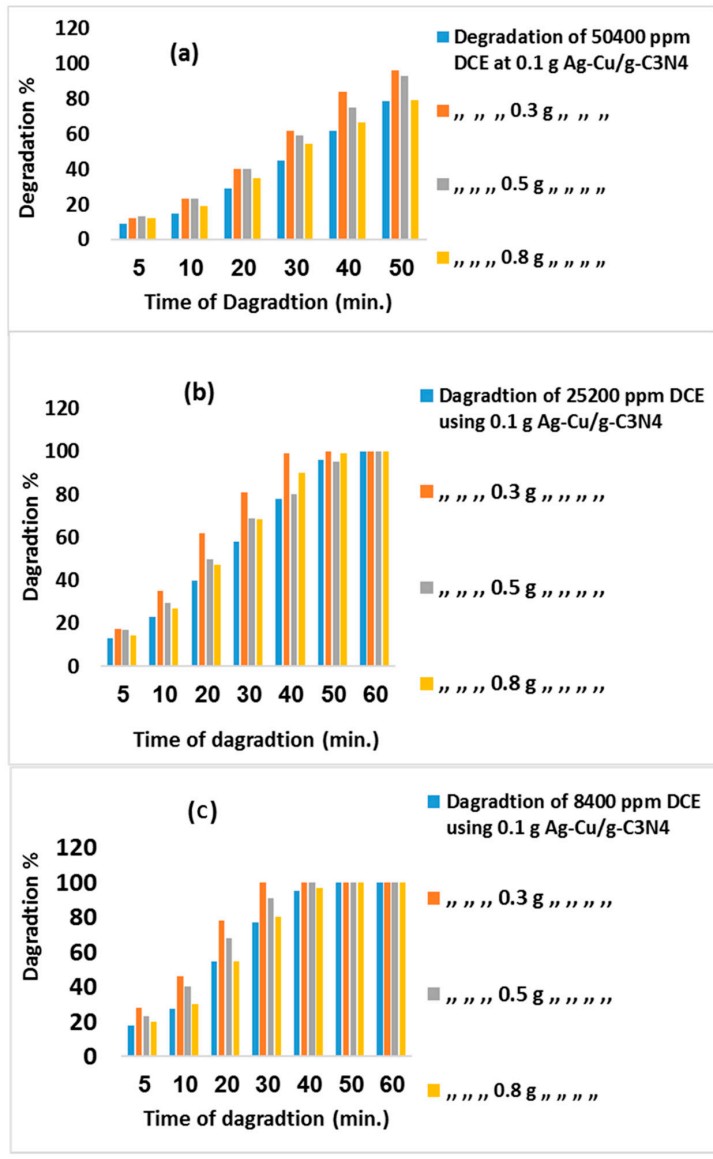

**Figure 9.** (**a**–**c**) Degradation % vs. time of degradation for different concentrations of (50,400, 25,200, and 8400 ppm) DCE using different doses of Ag-Cu/g-C$_3$N$_4$ (0.1, 0.3, 0.5, and 0.8 g/100 mL).

The efficiency of the sorption processes is highly influenced by the quantity of the sorbent and contact time. The effect of sorbent dosage was studied using 0.1, 0.3, 0.5, and 0.8 g Ag-Cu/g-C$_3$N$_4$ g/100 mL to remove 8400, 25,200, and 50,400 ppm DCE.

$$DCE\ removal\% = (C_0 - C_t)/C_0 \times 100 \tag{4}$$

where C$_0$ is the initial concentration and Ct (ppm) is the concentration at time t.

From Equation (4), it is possible to calculate the percentage of the DCE degradation in relation to the reaction time, the optimum dose for Ag-Cu/g-C$_3$N$_4$ to degrade the DCE, and the optimal concentration for DCE degradation.

From Figure 9a–c, a comparative survey of the photo-degradation results of DEC on binary Ag-Cu/g-C$_3$N$_4$ catalysts are summarized in Table 3, which shows that the degradation ppm of DCE on the Ag-Cu/g-C$_3$N$_4$ catalyst can reach as high as 100 degradations for 60 min at 0.3 and 0.5/100 mL at 50,400 ppm. In comparison, it reached 100% degradation at 25,200 and 8400 ppm dose concentrations, which is attributed to the low DCE concentration. The low catalyst dose (0.3, 0.5 g/100 mL) is faster than other doses for all different concentrations of 50,400 ppm of DCE, which is attributed to the Lowe steric hindered and soft coagulation of catalyst Ag-Cu/C$_3$N$_4$ at a low concentration. From Figure 9a–c, it is clear that the lower the concentration of DCE, the faster the degradation, and also the lower the concentration of Ag-Cu/g-C$_3$N$_4$, leading to the fast rate of degradation until a certain concentration (0.1 g/100 mL), due to a lower dose. The slow degradation at high concentrations of Ag-Cu/g-C$_3$N$_4$ is due to the effect of the steric hindrance and the start of the coagulation of molecules [45].

**Table 3.** Degradation% vs. time of degradation for different concentrations of DCE (a, 50,400 ppm), (b, 25,200 ppm), and (c, 8400 ppm) using different doses of Ag-Cu/g-C$_3$N$_4$ (0.1, 0.3, 0.5, and 0.8 g/100 mL).

| Time | Dose | 50,400 ppm | | | | 25,200 ppm | | | | 8400 ppm | | | |
|---|---|---|---|---|---|---|---|---|---|---|---|---|---|
| | | 0.1 Cat. * | 0.3 Cat. * | 0.5 Cat. * | 0.8 Cat. * | 0.1 Cat. * | 0.3 Cat. * | 0.5 Cat. * | 0.8 Cat. * | 0.1 Cat. * | 0.3 Cat. * | 0.5 Cat. * | 0.8 Cat. * |
| 5 | | 9.1 | 12.0 | 13.0 | 12.1 | 13.0 | 17.5 | 17.0 | 14.3 | 17.8 | 28.0 | 23.2 | 20.0 |
| 10 | | 15 | 23.0 | 23.0 | 19.0 | 23.0 | 35.0 | 29.5 | 27.0 | 27.4 | 45.9 | 40.0 | 30.0 |
| 20 | | 29.2 | 40.1 | 40.1 | 35.1 | 40.0 | 61.9 | 50.0 | 47.22 | 54.8 | 78.0 | 67.8 | 54.8 |
| 30 | | 45.0 | 62.0 | 59.1 | 54.2 | 58.2 | 80.9 | 69.0 | 68.25 | 77.0 | 100 | 91.1 | 80.0 |
| 40 | | 62.0 | 84.1 | 75.0 | 66.8 | 78.0 | 99.1 | 80.1 | 90.1 | 95.0 | 100 | 100 | 97.0 |
| 50 | | 79.0 | 96.0 | 93.0 | 79.2 | 96.0 | 100 | 95.2 | 99.0 | 100 | 100 | 100 | 100 |
| 60 | | 94.5 | 100 | 100 | 98.8 | 100 | 100 | 100 | 100 | 100 | 100 | 100 | 100 |

Cat. * is Ag-Cu/g-C$_3$N$_4$.

*3.6. Kinetics of the Process*

Adsorption kinetics are generally ruled using movie diffusion and intra-particle diffusion. However, the adsorption capability required for equilibrium attention has been investigated using the pseudo-first-order-kinetic reaction [50]. DCE molecules absorbed from the aqueous solution multiplied fast over time, and the equilibrium was finished within 60 min. The kinetic reaction has been proven in Figure 10 for being appropriately outfitted with the experimental information with the coefficients of determination (R2) of more than 0.8 [37,44].

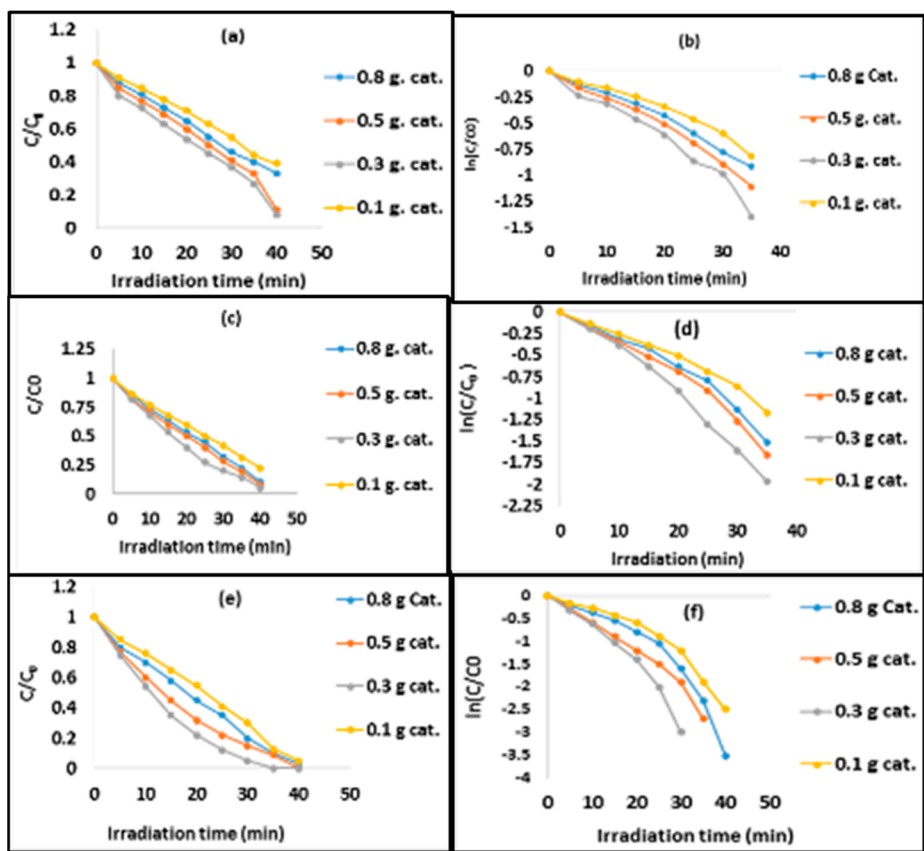

**Figure 10.** (**a**–**f**) The relation between irradiation time (min) and (C/Co) or ln(C/Co) for 50,400, 25,200, and 8400 ppm for DCE using different doses (0.1, 0.3, 0.5, and 0.8 g/100 mL) of Ag-Cu/g-C$_3$N$_4$.

A plot of ln(Co/C) vs. time is represented in Figure 10, which illustrates the rate of degradation of different concentrations (50,400, 25,200, and 8400 ppm DCE) at different doses (0.1, 0.3, 0.5, and 0.8 g) of Ag-Cu/C$_3$N$_4$. The values of k$_o$ and the linear regression coefficients of photo-degradation of the DCE, which correspond to different concentrations, are listed in Table 3.

The kinetic and regression constants (R$^2$) are presented in Table 4 and Figure 11, showing that the reaction rate constants (K$_{app}$) were increased by decreasing the dose of Ag-Cu/g-C$_3$N$_4$ (0.8 g to 0.3 g/100 mL) and returned to decreasing at 0.1 g/100 mL, which is logically due to increasing the degradation efficiency of different photocatalytic dosages (0.3, 0.5, 0.8, and decrease at 0.1 g) due to decreasing of the steric hinder of photons generated on Ag-Cu/g-C$_3$N$_4$ [45]. Figure 11 shows a higher rate of degradation for 8400 ppm DCE than 25,200 and 50,400 ppm, respectively, which is attributed to a decrease in the steric hindrance of the dichloroethylene molecule (DCE) in the low concentration (8400 < 25,200 < 50,400 ppm degradation). Figure 11 and Table 3 also confirm that the best dose of catalyst is 0.3 g/100 mL.

**Table 4.** Pseudo-first order apparent constant values for the different degradable DCE.

| DCE ppm | K$_o$ (min$^{-1}$) | | | | R$^2$ | | | |
|---|---|---|---|---|---|---|---|---|
| | Weight of Ag-Cu/g-C$_3$N$_4$ g/100 mL | | | | | | | |
| | 0.1 | 0.3 | 0.5 | 0.8 | 0.1 | 0.3 | 0.5 | 0.8 |
| 50,400 | 0.0233 | 0.049 | 0.0334 | 0.0274 | 0.9704 | 0.9322 | 0.9776 | 0.9867 |
| 25,200 | 0.0284 | 0.0597 | 0.0379 | 0.0337 | 0.9937 | 0.9532 | 0.9933 | 0.9907 |
| 8400 | 0.0422 | 0.0763 | 0.0607 | 0.0473 | 0.9886 | 0.9916 | 0.9993 | 0.9935 |

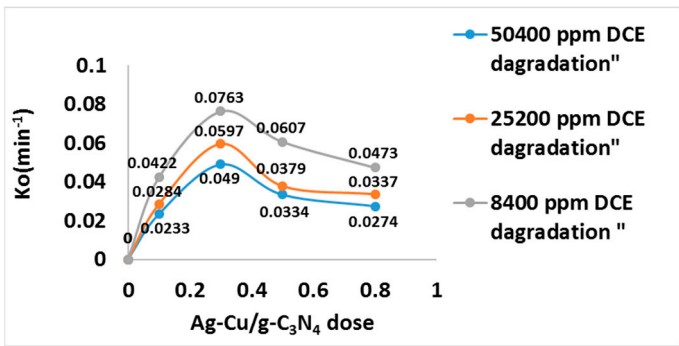

**Figure 11.** Pseudo-first order kinetic degradation of 50,400, 25,200, and 8400 ppm DCE using 0.1 g, 0.3 g, 0.5 gm, and 0.8 gm/100 mL, respectively.

On the basis of the previous work [49] and the above experimental results, a possible mechanism for the photocatalytic degradation DCE is proposed as follows: the electron–hole pairs can be quickly generated in the g-$C_3N_4$ upon UV-light excitation (Scheme 1) due to the SPR effect of bimetallic Ag-Cu nanoparticles, followed by the instant transfer of photo-generated electrons from the CB of g-$C_3N_4$ to the Ag-Cu nanoparticles at the interface of the Ag-Cu/g-$C_3N_4$ catalyst, which shifts the Fermi level to a more negative potential than the standard redox potential of $O_2/O_2^{\bullet-}$ ($-0.046$ V vs. NHE), thereby the dissolved oxygen can be readily reduced by negatively charged Ag nanoparticles to produce $O_2^{\bullet-}$ radicals [51,52]. Finally, the DCE molecules can be oxidized by $O_2^{\bullet-}$ radicals.

| g-$C_3N_4$ | + | hv | $\longrightarrow$ | g-$C_3N_4$ (h + e) |
|---|---|---|---|---|
| g-$C_3N_4$ (e) | + | Ag | $\longrightarrow$ | g-$C_3N_4$ + Ag(e) |
| g-$C_3N_4$ (e) | + | Cu | $\longrightarrow$ | g-$C_3N_4$ + Cu(e) |
| Ag(e) | + | $O_2$ | $\longrightarrow$ | $O_2^{\bullet-}$ + Ag |
| Cu(e) | + | $O_2$ | $\longrightarrow$ | $O_2^{\bullet-}$ + Cu |

**Scheme 1.** Possible photocatalytic mechanism of both g-$C_3N_4$, Ag, and Cu.

From the start, we indicated that the reaction proceeds through the adsorption–desorption mechanism and that the lower the concentration of DCE, the faster the degradation in time, and also the lower the concentration of the Ag-Cu/g-$C_3N_4$, the faster the degradation will be in order to reduce the steric hinder and coagulation of Ag-Cu/g-$C_3N_4$ and by comparing the reaction in the presence of visible light and the in the presence of UV-light irradiated.

Scheme 1 illustrates the photocatalytic mechanism of Ag-Cu/g-$C_3N_4$ composites during DCE decomposition under UV irradiation. Ag and Cu nanoparticle modification enhances the photocatalytic performance of g-$C_3N_4$ due to the synergistic effect of two aspects: one is the SPR effect of metals Ag and Cu, and another is the decrease in the recombination rate photo-generated $e^-$ $h^+$ pairs [52]. When Ag-Cu/g-$C_3N_4$ is irradiated by the simulated UV irradiation, the $e^-$ $h^+$ pairs are separated, $e^-$ is excited to CB of g-$C_3N_4$, and $h^+$ remains at VB of g-$C_3N_4$. Then, $e^-$ transfers to Ag and Cu NPs due to the high Schottky barrier of Ag and Cu, and finally, transfers to the photo-catalyst surface to join the reduction reaction. The generated $e^-$ comes from two routes, one from the plasmon-excited Ag and Cu NPs and the other from the photoexcited g-$C_3N_4$ nanosheets. This $e^-$ reacts with $O_2$ to generate $O_2^-$ radicals that can discompose DCE molecules into $CO_2$, HCl, and $H_2O$. Thus, it is concluded that the adsorbed Ag and Cu NPs have two functions: one is as the electron pool, and the other is the capture of the photoinduced electrons. As pictured in Figure 12A–C, an obvious absorption edge of Ag-Cu/g-$C_3N_4$

appears at about 550.6 nm, corresponding to the band gap of 2.7 eV, implying its UV-induced photocatalytic activity [16,53,54]. The strong and broad absorption band in the UV region for the Ag-Cu/g-C$_3$N$_4$ catalysts can be attributed to the localized SPR effect of metallic Ag and Cu nanoparticles, which shows efficient Plasmon resonance in the UV region [36,55]. Such an enhanced light absorption of the catalysts facilitates the yield of more electron–hole pairs, which subsequently results in a higher photocatalytic activity. Moreover, the SPR effect of metallic Ag and Cu nanoparticles causes the enhancement of the local electromagnetic fields, which speeds up the generation rate of photo-generated electron–hole pairs in the near-surface region of g-C$_3$N$_4$ [49,52,56,57]. The photo-generated electrons can be instantly scavenged by Ag and Cu nanoparticles at the interface of the Ag-Cu/g-C$_3$N$_4$ catalyst, creating a Schottky barrier that effectively reduces the probability of the recombination of photo-generated electron_hole pairs [42,43,58]. Figure 12A–C shows the PL emission spectra of the 50,400, 25,200, and 8400 ppm degradation using 0.3 g/100 mL Ag-Cu/g-C$_3$N$_4$ under the excitation wavelength of 550.6 nm. The strong emission peak of Ag-Cu/g-C$_3$N$_4$ centered at 550.6 nm suggests a high recombination probability of photo-generated electron–hole pairs.

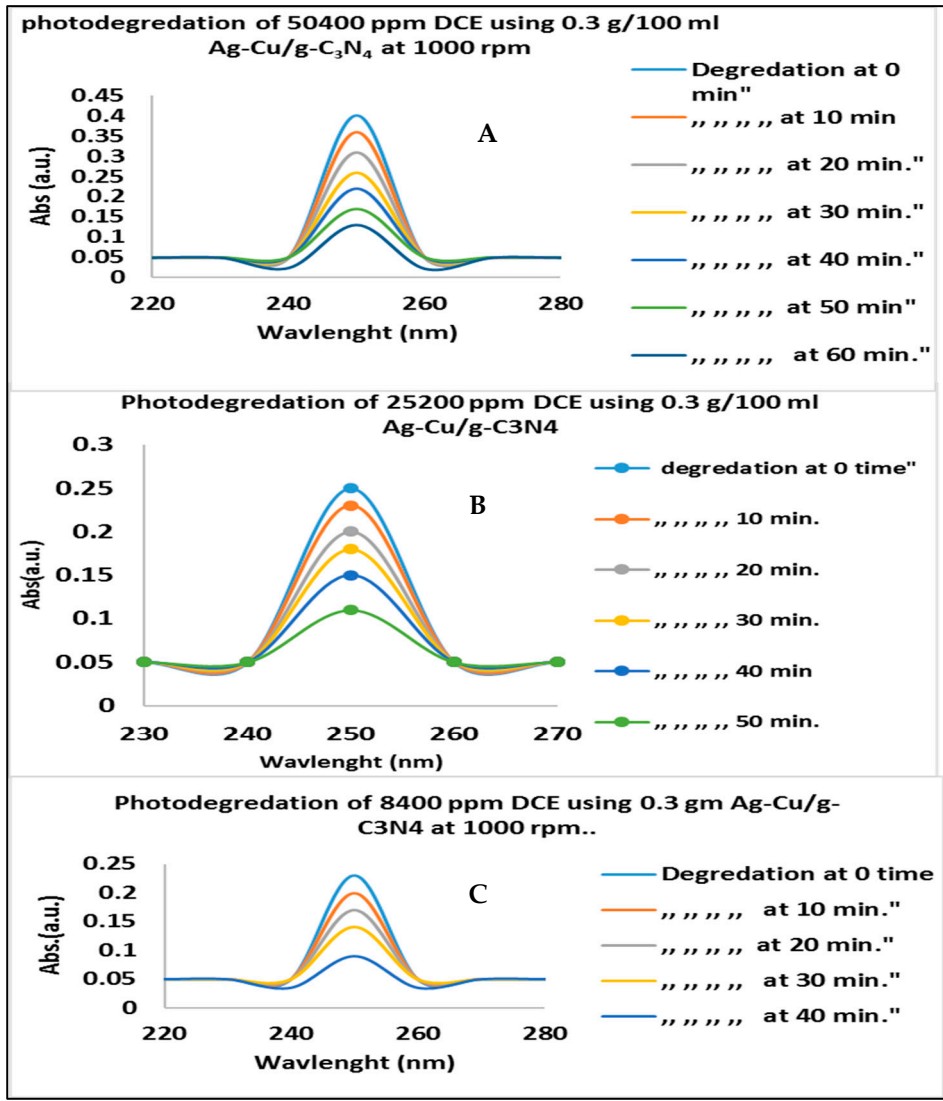

**Figure 12.** (**A–C**) Time-dependent UVVis absorption spectra for the degradation of 50,400, 25,200, and 8400 ppm DCE over 0.3 g/100 mL Ag-Cu/g-C$_3$N$_4$ [33].

Scheme 2 proposes a mechanism of the Ag-Cu/C$_3$N$_4$ photocatalytic degradation [49].

**Scheme 2.** Proposes a mechanism of the Ag-Cu/C$_3$N$_4$ photocatalytic degradation.

The decomposition occurs through the dehydrochlorination into vinyl chloride in the presence of OH$\cdot$ radical hydrolyzed by humidity alkaline OH and addition reaction occurs in the C = C bond to form intermediate ethyl chloride ion. This intermediate can be attacked by a nucleophilic oxygen species from the catalyst to form a chlorinated alkoxide species, which readily decomposes to gradually generate acetaldehyde, ace-tates, and CO$_2$ in addition to HCl [45,59–61].

Table 5 and Figure 13a–c show the result of the chloride ions released due to the degradation of DCE using different doses of Ag-Cu/g-C$_3$N$_4$, indicating a high concentration of Cl ions—equal to 0.3 g/100 mL at all DCE concentrations (50,400, 25,200, and 8400 ppm, respectively), which is the best low-dose catalyst due to low steric hindrance and poor Ag-Cu/g-C$_3$N$_4$ coagulation. Figure 13 also illustrates the clearance of different DCE concentrations (50,400, 25,200, and 8400 ppm) at different doses of 0.1, 0.3, 0.5, and 0.8 g photocatalytic Ag-Cu/g-C$_3$N$_4$ at 1000 rpm. The released Cl ion can be observed at different DCE concentrations of the order of 8400 and 25,200. The 50,400 ppm that initially affects the low concentration is broken down more than the high concentration. The Cl ion method determines a solution of chloride ions by titration with silver nitrate. A silver chloride precipitate forms as the silver nitrate solution is slowly added. The endpoint of the titration is reached when all the chloride ions have precipitated, as shown in Equation (5).

$$AgNO_3 + HCl \rightarrow AgCl \downarrow + NO_3^- \tag{5}$$

**Table 5.** Detection of liberated Cl ion resulted from the degradation of different concentrations of EDC molecules using different doses of photocatalytic Ag-Cu/g-C$_3$N$_4$ at 1000 rpm.

| Time of Degradn. | Cl Ion from Degraded 50,400 ppm DCE | | | | Cl Ion from Degraded 25,200 ppm DCE | | | | Cl Ion from Degraded 8400 ppm DCE | | | |
|---|---|---|---|---|---|---|---|---|---|---|---|---|
| | 0.1 * | 0.3 * | 0.5 * | 0.8 * | 0.1 * | 0.3 * | 0.5 * | 0.8 * | 0.1 * | 0.3 * | 0.5 * | 0.8 * |
| 0 | 0 | 0 | 0 | 0 | 0 | 0 | 0 | 0 | 0 | 0 | 0 | 0 |
| 10 | 60 | 100 | 80 | 50 | 98 | 130 | 100 | 75 | 120 | 160 | 150 | 125 |
| 20 | 100 | 150 | 130 | 79 | 180 | 242 | 223 | 140 | 160 | 314 | 260 | 225 |
| 30 | 145 | 185 | 165 | 131 | 270 | 350 | 312 | 230 | 253 | 432 | 390 | 324 |
| 40 | 180 | 230 | 200 | 170 | 365 | 450 | 403 | 308 | 371 | 570 | 502 | 443 |
| 50 | 235 | 270 | 250 | 212 | 400 | 500 | 455 | 360 | 463 | 700 | 633 | 573 |
| 60 | 272 | 310 | 290 | 250 | 420 | 523 | 465 | 390 | 500 | 765 | 690 | 607 |

* concentration of the Ag-Cu/g-C$_3$N$_4$ in 100 mL solution of the batch reactor.

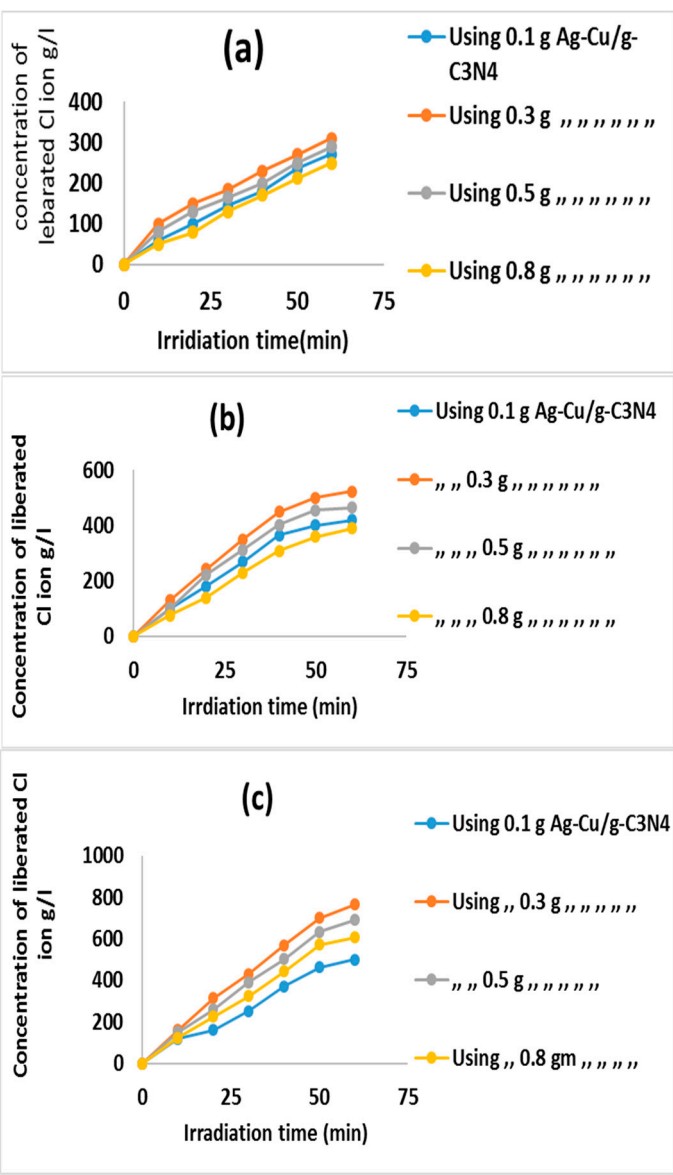

**Figure 13.** Detection of liberated Cl ion resulted from the degradation of different concentrations of (**a**) (50,400 ppm DCE degradation), (**b**) (25,200 ppm DCE degradation), and (**c**) (8400 ppm DCE degr adation) using different doses of Ag-Cu/g-$C_3N_4$.

## 4. Conclusions

In summary, the degradation of the residual DCE liquid released using the intermediate purification technique (DCE) is used to prepare the vinyl chloride monomer, which is then used to manufacture the polyvinyl chloride polymer. DCE is a poorly biodegradable and chemically stable environmental pollutant. Therefore, heterogeneous Ag-Cu/g-$C_3N_4$ photocatalysis was chosen as a nonselective method to degrade DCE via oxidation through reactions with reactive species (hydroxyl radicals and peroxy radicals) to HCl and $CO_2$. The kinetics and chloride ions released via the degradation of the DCE molecule can be detected via the titration of $AgNO_3$ using a potassium chromate indicator. Using a modified photocatalytic coupling-adsorption–desorption–oxidation method, the removal efficiency of these DCEs from the aqueous solution was confirmed. Using degradation, it was found that the overall performance increased with the increasing catalyst loading dose and decreasing overdose due to the increasing steric hindrance of the generated photon and initial nanocatalyst coagulation. The kinetic mechanism is mainly based entirely on hydroxyl radicals and $O_2^.$ offensive. Therefore, this liquid photocatalytic decomposition technology

of water-immiscible stable impurities and elements is capable of improving the quality of petrochemical industrial effluent.

**Author Contributions:** Conceptualization, E.G.B.; methodology, E.G.B., M.T., A.A.E., H.A.A., A.S.M. and F.S.; software, H.A.A. and F.S.; validation, E.G.B., M.T., A.A.E., H.A.A., A.S.M. and F.S.; formal analysis, H.A.A., A.S.M. and F.S.; investigation H.A.A., A.S.M. and F.S.; resources, E.G.B., M.T., A.A.E., H.A.A., A.S.M. and F.S.; data curation, M.T., H.A.A., A.S.M. and F.S.; writing—original draft preparation, H.A.A., A.S.M. and F.S.; writing—review and editing, E.G.B., M.T. and A.A.E.; visualization, E.G.B., M.T., A.A.E., H.A.A., A.S.M. and F.S.; supervision, E.G.B., M.T. and A.A.E.; project administration, E.G.B. and M.T.; funding acquisition, E.G.B. and A.A.E. All authors have read and agreed to the published version of the manuscript.

**Funding:** This research received no external funding.

**Institutional Review Board Statement:** The author's confirm that there were no ethical in preparing this manuscript.

**Informed Consent Statement:** This article does not contain any studies involving animal studies or human participants performed by any of the authors.

**Data Availability Statement:** Data supporting reported results can be found in the papers included in the References section.

**Acknowledgments:** This work was supported by the Petrochemical Department at the Faculty of Engineering, Pharos University, Chemical Department, Faculty of Science, Alexandria University, and the National Institute of Oceanography and Fisheries at Alexandria.

**Conflicts of Interest:** The authors declare no conflict of interest.

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
