# Peer review of "Photocatalytic Degradation of Chlorinated Hydrocarbons: The By-Product of the Petrochemical Industry Using Ag-Cu/Graphite Bimetallic Carbon Nitride"

_sustainability, doi:10.3390/su152216114_

Round 1

Reviewer 1 Report (Previous Reviewer 1)

Comments and Suggestions for Authors

Manuscript improved quality. It is acceptable.

Comments on the Quality of English Language

Minor editing of English language required

Author Response

First of all, the authors would like to express their great appreciation and thanks for the valuable and effective comments they have received from the editor and the reviewers. The manuscript has been revised accordingly.

Reviewer 1:

Minor editing of English

 Reply: Revised

Reviewer 2 Report (Previous Reviewer 3)

Comments and Suggestions for Authors

The revision is not up to the mark, following issues were detected.

Hence my recommendation is Major Revision

1. Where is Figure 1?

2. In the text Figure 2 is XRD but the figure 4 is XRD.

3. The first figure is starting from Figure 3????

4.  I could find the format related issues throughout the MS (fig, Figure, figure)

5. Figure 5 and 6: scale bars are not visible.

Author Response

  1. Where is Figure 1?

Reply: corrected

Figure 1 was added again from the supporting documents

Fig.1. Visually picture of graphitic carbon nitride and after doping with bimetallic silver and copper

The bimetallic catalysts were denoted Ag-Cu/ C3N4 (1: 1) with a yield of 0.8 gm as showing in Figure 1.

  1. In the text Figure 2 is XRD but the figure 4 is XRD.

Reply: mistake and it was corrected and moved to the XRD analysis region

Figure 4a, and b there are diffraction peaks at of 13.0° and 27.5° which may be listed to the (100) crystal plane bobbing up from the in-planar ordering of tris-triazine devices with a distance of 0.675 nm and (002) crystal plane of the stacking of the conjugated fragrant device with an interplanar distance of 0.326 nm, respectively. for Ag-Cu/g C3N4, there are observable peaks at of 38.1° and 44.3° which may be assigned to the (111) and (200) crystal planes of the face-focused cubic shape of Ag, respectively.

  1. The first figure is starting from Figure 3????

Reply: corrected

Figure 1 and 2 were added again from the supporting documents

Fig.1. Visually picture of graphitic carbon nitride and after doping with bimetallic silver and copper

The bimetallic catalysts were denoted Ag-Cu/ C3N4 (1: 1) with a yield of 0.8 gm as showing in Figure 1.

Fig.2. The phocatalytic reactor lab scale.

The change of concentration of DCE molecules is measured from the absorbance of the degradable DCE samples with a Unicam-9423-UV-E Spectrophotometer at its characteristic absorption peak wavelength of DCE at 550.6 nm. The photocatalytic degradation is illustrated in Figure 2.

  1. I could find the format related issues throughout the MS (fig, Figure, figure)

Reply: Corrected

  1. Figure 5 and 6: scale bars are not visible.

Reply: corrected

Figure 5a. Energy-dispersive X-ray (Sem-EDX) analysis of Ag-Cu/C3N4 and its elemental area mapping showing the distribution of elements

  • Figure 5b. Energy-dispersive X-ray (Sem-EDX) analysis of g- C3N4 and its elemental area mapping showing the distribution of elements.

a

b

Figure 6. (a) and (b)TEM photos of Ag-Cu/ C3N4.

Reviewer 3 Report (Previous Reviewer 4)

Comments and Suggestions for Authors

Accept in present form

Author Response

Accept in present form

Reply: Many thanks

Round 2

Reviewer 2 Report (Previous Reviewer 3)

Comments and Suggestions for Authors

Though revision has been done appropriately, I could find few format related issues, please fix them.

1. Table 1: space is needed after the word 'Table' and the number.

2. Check the reference number 21. 

3. Check figures 10 and 13, 14 and scheme 2 (to me the figures were overlaping on the same figure)

Author Response

Sustainability

Manuscript ID: sustainability-2622981

Manuscript Title: Photocatalytic degradation of chlorinated hydrocarbons. The by-product of the petrochemical industry using Ag-Cu/graphite bimetallic carbon nitride

Reply to reviewers’ comments

First of all, the authors would like to express their great appreciation and thanks for the valuable and effective comments they have received from the editor and the reviewers. The manuscript has been revised accordingly.

Follow updated notes from Reviewer 2

Though revision has been done appropriately, I could find few format related issues, please fix them.

  1. Table 1: space is needed after the word 'Table' and the number.

Reply:

Table 1.  comparison between the prepared material and the prepared previously reported works.

No.

Catalyst

Performance

reference

1

Degradation of 1,2-dichloroethane in  immobilized PAni-TiO2

The photocatalytic degradation of 1,2-DCE was about 60%, 90%, and 95% after 120 min, 240 min, and 300 min, respectively.

[12]

2

Degradation of 1,2-dichloroethane in a simulated wastewater solution: a comprehensive study by photocatalysis using TiO2 nanoparticles and zinc oxide

It was found that with the UV method just 55% of 1,2-DCA was removed after 6 h under 40 W UV radiation, but with the H2O2/UV method the removal reached 88% for a similar length of time and radiation intensity.

[38]

3

Degradation of 1,2-dichloroethane in V/TiO2

Complete photocatalytic degradation of 1,2-DCE was achieved after 120 min. by UV radiation

[7]

4

Degradation of 1,2-dichloroethane in UV-M lamp. UV-N/S2O42−

Complete degradation of 1,2-DCE after 300 min of irradiation time

[39]

5

Degradation of 1,2-dichloroethane in Fe/TiO2

The photocatalytic performance is a function of retention time and it would have a competitive adsorption on the active site of TiO2 between water vapor and 1,2-DCE.

[11]

6

Degradation of gaseous 1,2-dichloroethane using a hybrid cuprous oxide catalyst

degradation efficiencies of 83.8 and 82.2%

[40]

7

Reductive biodegradation of 1,2-dichloroethane by methanogenic granular sludge in lab-scale UASB reactors

1,2-DCE was converted mainly to ethane (65–80%) by The hydraulic retention time varied between 10 and 20 h.

[4]

8

Prepared material:

Photocatalytic degradation of 1,2 dichloroethane using Ag-Cu/graphite bimetallic carbon nitride radiated by UV

Degradation efficiencies of 0.3 g/100 ml Ag-Cu/g-C3N4 with a reaction time of less than 30 min of 100% with stable material and good reused several times;

- [Current research]

  1. Check the reference number 21.

Reply:

The reference talks about the photonic ability of substituted g-C3N4 to split water into hydrogen

21- HAYAT, Asif, et al. Visible-light enhanced photocatalytic performance of polypyrrole/g-C3N4 composites for water splitting to evolve H2 and pollutants degradation. Journal of Photochemistry and Photobiology A: Chemistry, 2019, 379: 88-98.

  1. Check figures 10 and 13, 14 and scheme 2 (to me the figures were overlaping on the same figure)

Reply: Corrected and Separated from each other

From Table 3. And Figure 10. a, b, and c. a comparative survey of the photo degrada-tion results of DEC on binary Ag-Cu/g-C3N4 catalysts are summarized in Table 2, which shows that the degradation ppm of DCE on Ag-Cu/g- C3N4 catalyst can reach as high as 100 degradations for 60 min at 0.3 and 0.5/100 ml at 50400 ppm. In compari-son, it reached to 100% degradation at 25200 and 8400 ppm dose concentration which is attributed to low DCE concentration. The low catalyst dose (0.3,0.5 g/100ml) is faster than other doses for all different concentrations of 50400 ppm of DCE which is at-tribute to Lowe steric hindered and soft coagulation of catalyst Ag-Cu/ C3N4 at low concentration. From Fig. 10(a, b & c), it is clear that the lower the concentration of DCE, the faster degradation, and also the lower the concentration of Ag-Cu/g- C3N4, leading to the fast rate of degradation till to a certain concentration (0.1 g/100 ml), due to more low dose. The slow degradation at high concentrations of Ag-Cu/g- C3N4, due to the effect of the steric hinder and start of coagulation of molecules [46].

Figure 10. a, b and c.  Degradation% vs time of degradation for different concentra-tion of (50400, 25200 and 8400 ppm) DCE using different dose of Ag-Cu/g-C3N4 (0.1, 0.3, 0.5 and 0.8 g/100 ml).

Scheme 2, proposed mechanism of the Ag-Cu/C3N4 photocatalytic degradation [50]

The decomposition occurs through by dehydrochlorination into vinyl chloride in the presence of e-. This intermediate can be attacked by nucleophilic oxygen species from the catalyst to form chlorinated alkoxide species, which readily decompose to gradually generate acetaldehyde, acetates and CO2 in addition to HCl [46,59-61]

Table 5. Detection of liberated Cl ion resulted from degradation of different concentration of EDC molecule using different dose of photocatalytic Ag-Cu/g-C3N4 at 1000 rpm.

        Cl ion conc.           and cat. Dos*.

Time of degradn.

Cl ion from degraded 50400 ppm DCE

Cl ion from degraded 25200 ppm DCE

Cl ion from degraded 8400     ppm DCE

0.1*

0.3*

0.5*

0.8*

0.1*

0.3*

0.5*

0.8*

0.1*

0.3*

0.5*

0.8*

0

0

0

0

0

0

0

0

0

0

0

0

0

10

60

100

80

50

98

130

100

75

120

160

150

125

20

100

150

130

79

180

242

223

140

160

314

260

225

30

145

185

165

131

270

350

312

230

253

432

390

324

40

180

230

200

170

365

450

403

308

371

570

502

443

50

235

270

250

212

400

500

455

360

463

700

633

573

60

272

310

290

250

420

523

465

390

500

765

690

607

* concentration of Ag-Cu/g-C3N4 in 100ml solution of batch reactor.

Table 5 and Figure 14 (a, b, c) show the result of chloride ions released due to DCE decomposition using different doses of Ag-Cu/g-C3N4, indicating a high concentration of chloride ions at an optimal catalyst concentration of 0.3 g/100 mL in all DCE con-centrations (50,400, 25,200, and 8,400 ppm, respectively), which is the best low-dose catalyst due to the low steric hindrance and poor coagulation of Ag-Cu/g-C3N4. Figure 14 also shows the removal of different DCE concentrations (50,400, 25,200, and 8,400 ppm) at different doses (0.1, 0.0.3, 0.5, and 0.8 g of Ag-Cu/g-C3N4 photocatalysis at 1000 rpm). The released Cl ion can be observed at different DCE concentrations of the order of 8400; 25200 >; The 50400 ppm that initially affects the low concentration is broken down more than the high concentration. The Cl ion method determines a solu-tion of chloride ions by titration with silver nitrate. A silver chloride precipitate forms as the silver nitrate solution is slowly added. The endpoint of the titration is reached when all the chloride ions have precipitated, as shown in Equation (5).

       AgNO3 +   HCl  →  AgCl    +   NO3            (5)

Figure 14. Detection of liberated Cl ion resulted from degradation of different concentrations of: [a (50400 ppm DCE degradation), b (25200 ppm DCE degradation), and c (8400 ppm DCE degradation) using different doses of Ag-Cu/g-C3N4

This manuscript is a resubmission of an earlier submission. The following is a list of the peer review reports and author responses from that submission.

Round 1

Reviewer 1 Report

Comments and Suggestions for Authors

1. “1. Introduction”

The research content and innovation of this paper are not reflected in the introduction, so it is suggested to supplement it.

 2. “Page 3”

“2.2 Laboratory preparation of the waste liquid (1, 2 dichloroethane [DCE])”, “2.3. Synthesis of g- C3N4 photo catalyst” and “2.4. Preparation of Ag-Cu/g- C3N4 Catalysts”

It is recommended to supplement the material preparation process with pictures.

 3. “Page 4, Figure 1”

It is suggested to change the title to graphite carbon nitride before and after doping with bimetallic silver and copper.

 4. “Page 7, Figure 6”

It is recommended that the comments of Figure 6 be marked clearly.

 5. “Page 8, Line 273”

“… take a shorter time as show Figure 12(a, b, c and d).”

There is no a, b, c and d in Figure 12, please check it.

 6. “Page 9, Line 312”

“A comparative survey of the photodegradation results of DEC on binary Ag-Cu/g-C3N4 catalysts are summarized in Table 2, which shows that the degradation % of DCE on Ag-Cu/g-C3N4 catalyst can reach as high as 100% gradation % Under UV light irradiation for 60 min at all dose concentration (0.1-0.8 g/100 ml) DCE.”

The text does not match the description in Table 2, please check it. It is also recommended to check whether this statement is incorrect.

 7. “Page 10, Figure 11”

No explanation of Figure 11 can be found in the article, it is suggested that a description of Figure 11 needs to be added.

 8. “Table 1, Table 3 and Table 4”

“EDC” “DEC”.

 9. “Page 13, Figure 13”

It is recommended to indicate the concentration of DEC in Figure13 (a, b and c) respectively.

Comments on the Quality of English Language

Moderate editing of English language required

Reviewer 2 Report

Comments and Suggestions for Authors

Authors report Environmentally friendly synthesis of bimetallic Ag-Cu/graphitic carbon nitride for photocatalytic degradation of chlorinated hydrocarbon Side product liberated from the petrochemical industry. Manuscript is poorly written with roughly designed figure of substantially low resolution. Moreover, results are not discussed well. Therefore, I don’t recommend this manuscript for publication unless revised significantly. Following are some comments for the authors.

1.     The title is too long and wordy, authors are suggested to keep the title as short as possible while still conveying the main idea.

2.     The abstract is poorly written and mismanaged, which doesn’t provide any solid idea about the work. The abstract should cover the key elements of the work, a clear statement of the research objective or problem and results in a concise way.

3.     Authors are suggested to focus on the primary research question and their approach, signifying the novelty of this work.

4.     Results ‘’This section may be divided by subheadings. It should provide a concise and precise description of the experimental results, their interpretation, as well as the experimental conclusions that can be drawn.’’ Authors forgot to remove the instruction in the manuscript template. Please revise the manuscript carefully before submitting.

5.     Authors are suggested to redesign the figures for better presentation; for instance, minimize the figures by merging some important figures and moving others to supplementary information. Figures quality is very poor and doesn’t meet the publication standards, for instance, Fig 13 is so blur and can’t be read.

6.     The manuscript should be revised carefully to refine the language for clarity and coherence.

7.     Results are not discussed well and significantly lack the scientific insights behind the analysis. Authors should briefly discuss each analysis and results with respect to published literature or scientific background.

8.     Figure 10, the degradation time is not obvious, whether it is in minutes or hours.

9.     The references should be updated with recent literature, such as Catalysts 2023, 13(2), 231; J. Catal. 421, 221 (2023), Battery Energy. 2023, 20220060

Comments on the Quality of English Language

Must be improved 

Reviewer 3 Report

Comments and Suggestions for Authors

Blall et. al reported the synthesis and characterization of Ag and Cu bimetals incorporated g-C3N4 material (Ag-Cu/g-C3N4) for the degradation of 1, 2-dichloroethane (DCE). The results are promising and interesting. But the work needs more in-depth studies to enhance the understandability and novelty.  The novelty of this work can be enhanced by adding more characterization and different photodegradation studies. This article is not acceptable for publication in its current form.

Recommendation: Reject and Resubmit.

There are a few concerns and comments for the further improvement of the manuscript, they are listed below,

1.      The introduction should present the research problem, provide background information about the topic, and highlights the significance and purpose of the study.

2.      Please consider the punctuation and grammar in the manuscript.

3.      The quality of the graphs should be enhanced for a better understanding to the readers.

4.      The authors stated “The result also indicates 260 the incident wavelength light on the catalyst is small than the bandgap of both Ag, Cu 261 nanoparticles (2.51 eV), (1.98-2.02eV.) and bandgap of g-C3N4 (2.7–2.8 eV.)”, please provide a suitable citation for the mentioned data.

5.      The energy band gap of the synthesised catalyst can be obtained by Differential Reflectance Spectroscopy (DRS), please mention the band gap energy of the material in the manuscript. Please refer and cite https://doi.org/10.1038/s41598-019-52191-9 .

6.      The surface area of the material is an important factor for the catalyst, please provide the BET surface area of the prepared material. Please refer and cite https://doi.org/10.1016/j.cej.2023.141577 .

7.      Authors should mention the comparison between silver (Ag), Copper (Cu) metals, g-C3N4 and prepared Ag-Cu/C3N4 materials to enhance the novelty of the material.

8.      Information regarding the calibration curve can be shifted to supplementary information.

9.      Better to explain equations and methods in the ‘methods section’ rather than in the results and discussion.

10.  Figure 12 and Table 3 represent the best dose of catalyst is 0.3 g/100ml. Please include the reason behind the result.

11.  Mechanism of photocatalytic degradation can be predicted by the methods such as radical scavengers’ activity studies.

12.  What is the significance of the result of liberated chloride ion resulting from the degradation of DCE using different dos of Ag-Cu/g-C3N4. What is the conclusion of the results obtained by these experiments?

13.  pH of the solution significantly varies the efficiency of degradation. Better to mention the effect of pH in the manuscript.

14.  In a real-time application, the concentration of salts alters the degradation efficiency. Please perform the effect of salt concentration studies.

15.  The reusability of the catalyst is an important parameter to study. The reusable properties of the catalyst are an important factor in industrial applications, please provide more details.

16.  The degradation of DCE might produce secondary pollutants, the authors can perform GC-MS analysis for a better understanding. Please refer and cite https://doi.org/10.1016/j.apsusc.2020.146974 .

17.  Better the authors express the concentration of DCE in ppm or mM for better understanding.

18.  Please include a table showing a comparison between the prepared material and the previously reported works.

Comments on the Quality of English Language

Please fix all grammatical errors in the MS

Reviewer 4 Report

Comments and Suggestions for Authors

1, 2-Dichloroethane (DCE) is one of the maximum critical chlorinated unstable natural pollutants in waste liquid. The authors optimized and modified g-C3N4 by doping Ag and Cu bimetals which alter the photochemical properties of g-C3N4, narrowed the band gap, expanded photoabsorption into the visible range, and improved the photocatalytic performance quantum efficiencies. However, some important results and conclusions should be clarified clearly before considering for publication.

1.      Some spelling mistakes should be revised. For example, g-C3N4 (Line 212, page 5) in which 3 and 4 should be in subscript etc..

2.      Some figures in your paper are very blurry. Please consider replacing them with clearer ones. Figure 1, Figure 13 and etc..

3.      For the degradation processing of 1, 2-Dichloroethane, what is the effect of carbon nitride?

4.      As mentioned in the introduction, TiO2 has many defects. What is the effect of the improved catalyst under ultraviolet conditions compared with TiO2?

5.      The free radical trapping experiments should be supplemented.